# Forecasting the monthly incidence rate of brucellosis in west of Iran using time series and data mining from 2010 to 2019

Hadi Bagheri[1], Leili Tapak[2,3], Manoochehr Karami[1,3], Zahra Hosseinkhani[4], Hamidreza Najari[4], Safdar Karimi[5], Zahra Cheraghi[1,3] *

1 Department of Epidemiology, School of Public Health, Hamadan University of Medical Sciences, Hamadan, Iran, 2 Department of Biostatistics, School of Public Health, Hamadan University of Medical Sciences, Hamadan, Iran, 3 Modeling of Noncommunicable Diseases Research Center, Hamadan University of Medical Sciences, Hamadan, Iran, 4 Department of Health Services Management, School of Health, Qazvin University of Medical Sciences, Qazvin, Iran, 5 Department of Prevention and Fighting of Diseases of Deputy of Health of Qazvin University of Medical Sciences and Health Services, Qazvin, Iran

* cheraghiz@ymail.com

**Data Availability Statement:** All relevant data are within the paper and its Supporting Information files.

## Abstract

### Background

The identification of statistical models for the accurate forecast and timely determination of the outbreak of infectious diseases is very important for the healthcare system. Thus, this study was conducted to assess and compare the performance of four machine-learning methods in modeling and forecasting brucellosis time series data based on climatic parameters.

### Methods

In this cohort study, human brucellosis cases and climatic parameters were analyzed on a monthly basis for the Qazvin province–located in northwestern Iran- over a period of 9 years (2010–2018). The data were classified into two subsets of education (80%) and testing (20%). Artificial neural network methods (radial basis function and multilayer perceptron), support vector machine and random forest were fitted to each set. Performance analysis of the models were done using the Root Mean Square Error (RMSE), Mean Absolute Error (MAE), Mean Absolute Root Error (MARE), and $R^2$ criteria.

### Results

The incidence rate of the brucellosis in Qazvin province was 27.43 per 100,000 during 2010–2019. Based on our results, the values of the RMSE (0.22), MAE (0.175), MARE (0.007) criteria were smaller for the multilayer perceptron neural network than their values in the other three models. Moreover, the $R^2$ (0.99) value was bigger in this model. Therefore, the multilayer perceptron neural network exhibited better performance in forecasting the studied data. The average wind speed and mean temperature were the most effective climatic parameters in the incidence of this disease.

**Funding:** This study was funded by the Vice-Chancellor of Research and Technology of Hamadan University of Medical Sciences. The funders had no role in study design, data collection and analysis, decision to publish, or preparation of the manuscript.

**Competing interests:** The authors have declared that no competing interests exist.

## Conclusions

The multilayer perceptron neural network can be used as an effective method in detecting the behavioral trend of brucellosis over time. Nevertheless, further studies focusing on the application and comparison of these methods are needed to detect the most appropriate forecast method for this disease.

## Background

Brucellosis (Malta fever) is one of the most common zoonotic diseases and has long been one of the most important health concerns for humans and animals since old times [1]. The significance of this disease is not limited to its physical complications, and is one of the most important challenges of economic development in many countries–including Iran- as economic development in Iran still depends on its agriculture and ranching [1–3]. Direct contact with the infected livestock or dairy products is one of the most common routes of transmission, although the main transmission route is through the consumption of raw milk and other unpasteurized dairy products [4–6]. The prevalence of brucellosis is globally widespread, nevertheless, the highest prevalence is seen in the Mediterranean region, Arabian Peninsula, Indian Subcontinent and parts of South and Central Americas [7–9]. This disease still persists as an undetectable endemic disease in many developing countries [10,11]. According to the World Health Organization (WHO), annually, 500000 cases of infection are reported globally, and for every detected case, four cases go undetected [12–15]. Although brucellosis has been eradicated in many industrial countries, it is still a serious health threat in some countries, including Iran [16–18]. In Iran, brucellosis is recognized as an endemic disease that is annually reported in the northern and north-western parts of the country at high rates, and will lead to extensive problems [17]. To reduce the rate of this disease, and prevent its associated problems, strategic planning must be done and control and prevention measures must be taken based on applied management by health officials and planners. To this end, the utilization of modelling techniques seems necessary for the timely detection of the epidemic in the future and the early detection of the changing trend of the disease over time. These must be done for the timely and appropriate execution of control measures such as sensitization and education of physicians on the diagnosis and treatment of these patients, and delivery of health messages regarding prevention, etc.

To achieve this goal, quality data and forecast methods with the least errors are required [19]. The healthcare system is an important means of collection, analysis, interpretation and dissemination of healthcare data results, which is mainly used to prevent and control diseases and health events [20]. The health system has been designed to facilitate the detection of abnormal behavior of infectious diseases and other health events. To achieve this goal, different statistical methods have been used to forecast infectious diseases. Time series models have been used by researchers for a long time now. They attempt to forecast the epidemiologic behavior of diseases using historical surveillance data. In the past, researchers have used various time series models to forecast the incidence of epidemics, such as, exponential smoothing [21], generalized regression [22], analysis [23], and multilayered time series models [24]. However, the use of these models requires the determination of exact mathematical parameters and the establishment of underlying hypotheses, particularly the linearity of the regression association [25]. In recent years, time series models based on machine learning methods–such as the artificial neural network- have been used to model the time series incidence of infectious

diseases [26]. It has been demonstrated that these methods are effectively better at forecast than the classic methods. The artificial neural network (ANN) is a powerful non-linear technique used in data modeling that can model the complex connections between forecasting variables and the target without taking into account any primary hypothesis and previous knowledge of the relations between the parameters under study [27]. Two pioneer methods in neural networks are the Radial Basis Function (RBF) and the Multilayer Perceptron (MLP) networks. RBF is a more common type of neural network learning which responds to a limited section of the input space; it has a faster and more accurate and yet simpler network structure compared to other neural networks, while the MLP is more generalizable [28]. Another machine learning method is the Support Vector Machine (SVM) method. SVM is a macro data method that is used owing to its desirable performance in regression problems and classification when compared to classic models. This model employs a risk function including empirical error and a regularization principle [29]. It has higher power and better performance in practical applications. This trait is due to its structural principle of risk minimization; it has greater generalizability and is superior to the empirical risk minimization principle. SVM have been employed in different time series problems, namely, machinery industry [30], engine reliability prediction/forecast [31] and forecasting economics time series [32,33]. SVM success in forecasting time series of different fields of science led us to the conclusion that we should use it for forecasting brucellosis time series. Many researchers have approved the desirable performance of these four techniques and their advantages in forecast [28]. Nonetheless, in spite of the widespread application of these techniques, they have not–to our knowledge- been evaluated for Qazvin's brucellosis data. The precise and timely forecast of trend changes in outbreak control management are very important, and the performance of various methods depend on the data, and their performance may differ for different data. Therefore, the goals of this study were to assess the performance of artificial neural networks (including, the RBF and MLP–separately), the SVM and random forest in forecasting the number of brucellosis cases and to identify a model with better forecast abilities. This model may then be utilized in the public health system, to control and prevent the high incidence of brucellosis.

From the climatic perspective, it is essential to determine the epidemiologic conditions of brucellosis in terms of environmental circumstances for different regions. This in turn demands the examination of environmental factors of each region. Of the most significant environmental characteristics of each region are its climatic and weather conditions and other influential factors. Given the bacterial nature of brucellosis, detecting climatic/weather characteristics and other influential factors can greatly help manage and control this disease. Hence, the other goal of this study was to determine the impact of climatic factors such as, Average temperature, minimum and maximum temperature, precipitation, wind speed and average wind speed, and other variables, such as, mean age, gender ratio, rural ratio, ratio of unpasteurized dairy product consumption, and contact with livestock on the incidence of brucellosis–using machine learning methods. Thus, by determining the most appropriate model, the results of this research can prove beneficial to epidemiologists in preventing and controlling epidemics.

## Methods

### The data and area under study

This study was conducted on time series data of brucellosis using the following covariates: month, season, year, rural ratio, mean age, males ratio, ranchers' ratio, ratio of contact with livestock, ratio of consumption of unpasteurized dairy products, and climatic parameters, including, Average temperature, minimum and maximum temperature, precipitation, wind

speed and average wind speed in Qazvin province–on a monthly basis. Qazvin is located in North-western Iran and at the southern skirts of the Alborz Mountain Range. It is cool in summer and cold in winter. There is an appropriate distribution of humidity across Qazvin due to the effect of rain-producing air masses and altitudes. The trend of humidity changes during the year indicates maximum humidity during winter and minimum humidity during summer. Based on the most recent national geographical divisions made by the Ministry of Interior in 2013, Qazvin province has an area of 15567 m$^2$, and includes 6 counties, Qazvin, Buin-Zahra, Abyek, Avaj, Takestan and Alborz.

Based on national guidelines, the patients' clinical and epidemiological data are registered online in the Health Surveillance System. Accordingly, patients with the following clinical–epidemiological symptoms of brucellosis were considered disease cases: fever, myalgia and para-clinical symptoms (the results of two routine lab tests for brucellosis) including, Wright's (diagnostic test for brucellosis; values greater than 1.8 indicate presence of infection) and 2ME (Mercaptoethanol Brucella agglutination test) (brucellosis confirmatory test, which if greater than or equal to 1.4 is indicative of the presence of infection) [34,35].

Here the trend of a number of human brucellosis cases was analyzed using some covariates and monthly climatic parameters during 2010–2018 in Qazvin province. Data on the number of brucellosis cases and covariates (including, rural ratio, mean age, gender ratio, ratio of contact with livestock, ratio of unpasteurized dairy product consumption) were extracted from the databank of Qazvin University of Medical Sciences' Deputy of Health, and data related to climatic parameters were obtained from Qazvin province's Meteorological System. To examine the validity of the models applied in this study, the monthly data were classified into two sets, the training and test sets. This classification was done based on the performance assessment of time series data. Studies conducted on time series data consider a 70 or 80 percent ratio of data as the training set of data (from the beginning of the series) and the remainder are considered as the test set [36,37]. Therefore, here too, the 80 to 20 percent ratio was considered for the data as the training (from April 2010 until August 2017) and the test (from September 2017 until March 2018) sets, respectively.

**Models.** In this study, four machine learning methods including the radial basis function, multilayer perceptron, support vector machine and Random forest time series were employed to forecast monthly changes of brucellosis frequency using covariates and climatic parameters. Auto Regressive Integrated Moving Average (ARIMA) was fitted to the data with 1–12 lags for the monthly brucellosis data, covariates and climatic parameters. A significance level of 0.05 was taken into consideration.

**Support vector machine.** The SVM is a machine learning method that is used due to its desirable performance in regression and classification problems compared to the classic models. This model employs a loss function including empirical error and a regularization principle [29]. When dealing with regression problems, this method attempts to estimate the relationship between response variable and covariates using a linear function in a higher dimension instead of a non-linear function in the initial space of data Suppose *y(t)* is a set of time series data that depends on time $t \in \{1, 2, \ldots n\}$. In time series problems, the goal is to create a forecast rule based on current and past data that can be used to estimate future values. Therefore, the function *f(.)* is defined as a function that reverses an output to forecast future values [29]. The following equation is a forecast function for non-linear regression:

$$f(y) = w.\phi(y) + b \qquad (1)$$

The SVM depicts the data that are nonlinear in their input space in a higher dimensional feature space through the kernel function$\phi(.)$, which must be accurately selected. Therefore, a

linear problem will be obtained. In order to estimate the forecast rule, the (weights) $w$ coefficient and x-intercept $b$ must be optimized.

There are a number of different kernels [38]. In our study, the kernel function was used with better performance upon examining different kernel functions' performances.

*Artificial neural networks.* Artificial neural networks (ANN) are data processing mathematical tools used in many scientific fields for forecasting, pattern recognition and classification [36]. There are several nodes and weights that connect the nodes to each other. Several ANNs exist, of which the MLP and RBF have been applied in many studies and been compared with each other. The MLP is a special type of this method which has non-linear activation functions such as the sigmoid in the hidden layer and the linear function in the external layer [36]. The relationship between the input and hidden layers was is as below:

$$y_j = f\left(\sum_{i=1}^{N} w_{ji} x_i + b_j\right) \tag{2}$$

in which, $x$ is the nodal value of the previous layer, $y$ is the nodal value of the current layer, $b$ is the intercept of the current layer, and $w$ represents the regression coefficients or weights [39,40];

To fit the MLP model, two hidden layers, one input and output layers were used in this study. Sigmoid and tangent hyperbolic functions were considered in the hidden layers and identity function was used in the output layer.

*Random forest.* The Random forest (RA) technique is a regression and classification tool based on a set of tree forecasters [41]. For a regression problem, RA combines the forecasts obtained from several regression trees, such that, each tree is built by splitting down the predictor space return. (Analysis continues up to the point that the constructed sub-spaces become homogenous and similar) [42]. The RA algorithm includes, 1) the stage of extracting many bootstrap samples from the primary data and construction of training sets, 2) growing a regression tree for each of the train samples obtained, 3) finally, predicting the response variable for the new data by accumulating the predictions obtained from all trees [43].

*Model assessment criteria.* To assess and compare the accuracy of prediction and the performance of the models in the times series data modeling in this study, the Root Mean Square Error (RMSE), Mean Absolute Error (MAE) and Mean Absolute Root Error (MARE), and $R^2$ determination coefficient criteria were used, which were calculated by the following relations [28,44]:

$$RMSE = \sqrt{\frac{1}{n}\sum\left(Y_{obs} - Y_{pred}\right)^2} \tag{3}$$

$$MAE = \frac{1}{n}\sum\left|Y_{obs} - Y_{pred}\right| \tag{4}$$

$$R^2 = 1 - \frac{\sum\left(Y_{obs} - MY_{pred}\right)^2}{\sum\left(Y_{obs} - \bar{Y}\right)^2} \tag{5}$$

$$MARE = \frac{1}{n}\sum\frac{\left|Y_{obs} - Y_{pred}\right|}{Y_{obs}} \tag{6}$$

In the associations/relations above, $Y_{obs}$ and $Y_{pred}$, respectively, represent the numbers of brucellosis cases observed and predicted.

**Table 1. Descriptive characteristics of brucellosis cases in Qazvin Province.**

| Variables | Category | Frequency (percent) | P-values |
|---|---|---|---|
| **Age Group** | 0–9 years | 163 (5.10) | 0.026 |
| | 10–19 years | 329 (10.30) | |
| | 20–29 years | 654 (20.47) | |
| | 30–39 years | 653 (20.44) | |
| | 40–49 years | 486 (15.21) | |
| | 50–59 years | 436 (13.65) | |
| | ≥60 years | 473 (14.80) | |
| **Gender** | Male | 2026(63.43) | <0001 |
| | Female | 1168(36.57) | |
| **Contact with livestock** | Yes | 2452 (76.76) | <0001 |
| | No | 742 (23.23) | |
| **Habitat** | Urban | 1113(34.84) | <0001 |
| | Rural | 2081(65.15) | |
| **Job type** | Housewife | 982 (30.75) | <0001 |
| | Rancher—Farmer | 1287 (40.29) | |
| | Student | 236 (7.38) | |
| | Employee | 49 (1.53) | |
| | Worker | 139 (4.35) | |
| | Private | 144 (4.50) | |
| | Others | 357 (11.17) | |
| **Consumptions of unpasteurized dairies** | Yes | 2612 (81.77) | <0001 |
| | No | 582 (22.18) | |

*Implementation and parameter tuning.* To implement the models, variables in Table 1 as well as climatic variables of wind speed (m/s) and temperature (Centigrade) were used as predictors and the numbers of brucellosis cases observed was used as the output. Then, all the three machine learning techniques of RF, SVM and ANN were implemented to predict. For all the three models, there were some parameters to be tuned. To this, first, we divided the data set into two sets of training and testing (80–20%). Then, we conducted a 10-fold cross-validation over the training set to find the optimum values. For the SVM, two parameters of C and gamma were tuned and the optimum values obtained were 0.023 and 0.008, respectively. For the ANN, the number of hidden layers needed to be selected using cross-validation. So, we considered 1–3 hidden layers and an ANN with two hidden layers was selected as the optimum. This was the case for both MLP and RBF. For the random forest, the number of trees and mtry (the number of covariates randomly selected from all predictors to create each tree) were tuned and a RF with 550 trees and mtry = 3 was selected as the optimum parameters. The models were trained using the data in the training set and were tested on the testing set.

**Software.** All analysis was done using, R 3.4.2 in fitting the models, covariates and dependent variables were normalized–which was done with the equation below [44]:

$$Y_{Normalized} = \frac{Y - Y_{min}}{Y_{max} - Y_{min}} \tag{7}$$

## Results

The examination of data of 3194 registered brucellosis patients showed that between the years of the study (2010–2018) most patients (63.4%) were males and the remaining were females (Table 1). Their mean age was 38.43±10.28 years; after classifying the individuals into 5-year

**Table 2. Frequency of brucellosis cases by year and season in Qazvin Province.**

| Year | Spring | Summer | Autumn | Winter | P_value | Total |
|---|---|---|---|---|---|---|
| | Number (Percent) | Number (Percent) | Number (Percent) | Number (Percent) | 0.166 | Number (Percent) |
| 2010 | 51 (24.56) | 69 (35.94) | 37 (19.27) | 35 (18.23) | 0.488 | 192 (6.01) |
| 2011 | 85 (27.16) | 87 (27.80) | 64 (20.45) | 77 (60.24) | 0.299 | 313 (9.79) |
| 2012 | 82 (25.87) | 109 (34.38) | 59 (18.61) | 67 (21.14) | 1.00 | 317 (9.92) |
| 2013 | 72 (24.08) | 90 (30.10) | 56 (18.73) | 81 (27.09) | 1.00 | 299 (9.36) |
| 2014 | 101 (22.54) | 149 (33.26) | 81 (18.08) | 117 (26.12) | 0.488 | 448 (14.02) |
| 2015 | 118 (23.05) | 153 (29.88) | 83 (16.21) | 158 (30.89) | 0.166 | 512 (16.03) |
| 2016 | 119 (28.95) | 120 (29.20) | 88 (21.41) | 84 (20.44) | 0.083 | 411 (12.86) |
| 2017 | 94 (26.93) | 92 (26.36) | 83 (23.78) | 80 (22.92) | 0.166 | 349 (10.92) |
| 2018 | 69 (19.55) | 92 (26.06) | 86 (24.36) | 106 (30.03) | 1.00 | 353 (11.05) |
| Total | 791 (24.76) | 961 (30.08) | 637 (19.94) | 805 (25.20) | 0.634 | 3194 (%100) |

age groups we observed that the highest percentage of the disease had occurred in the third and fourth decades of life, i.e. between 25 to 39 years (31.18%) [Table 1]. The examination of employment status revealed that the most commonly affected job was ranching–farming (40.29%) [Table 1]. Upon examining the status of the disease per rural and urban regions, the highest frequency was seen in rural regions, at a rate of 65.24% [Table 1]. Upon examining the probable risk factors of this disease, the consumption of unpasteurized dairy products (81.77%) and contact with livestock (76.76%) had the highest frequencies [Table 1]. We examined the monthly pattern of the disease, and found that April (6.32%) and August (10.49%) had witnessed the lowest and highest percentages of disease, respectively [Table 2]. Regarding the seasonal pattern of disease, the highest and lowest percentage frequencies were seen in summer (30.08%) and autumn (19.94%), respectively [Table 2]. The year 2015 (16.03%) witnessed the highest reporting rate of the disease among the years of the study. Moreover, the lowest frequency percentage was reported in 2010 (6.01%) [Table 2]. The mean 9-year incidence weight of the disease for each of Qazvin's counties indicated that Avaj (222.42 per 100000 person) and Takestan (42.63 per 100000 person) held first and second positions, respectively, while the provincial incidence was 27.43 per 100,000 person [Table 3, Fig 1]. We also extracted the statistical features of climatic parameters, the results of which are as follows; mean temperature: 14.59±9.05, precipitation: 25.62±24.02, wind speed: 1.88 ±34, maximum temperature: 28.21±09.63, minimum temperature: 1.59±8.52, wind speed: 14.36 ±04.09 (Table 4).

**Table 3. Annual brucellosis incidence rates (per 100,000) by counties of Qazvin Province.**

| Year | Buin Zahra | Alborz | Abyek | Takestan | Qazvin | Avaj | County |
|---|---|---|---|---|---|---|---|
| 2010 | 31.57 | 5.41 | 28.77 | 24.28 | 10.59 | - | 15.97 |
| 2011 | 43.71 | 13.77 | 37.30 | 21.39 | 24.88 | - | 26.04 |
| 2012 | 47.23 | 9.94 | 47.88 | 23.14 | 23.22 | - | 26.06 |
| 2013 | 11.48 | 2.84 | 16.95 | 23.08 | 13.26 | - | 12.88 |
| 2014 | 23.00 | 11.01 | 45.62 | 83.35 | 14.36 | 283.12 | 35.99 |
| 2015 | 59.81 | 19.15 | 33.90 | 60.22 | 13.88 | 400.90 | 40.67 |
| 2016 | 36.78 | 15.30 | 33.79 | 56.65 | 14.72 | 255.65 | 32.60 |
| 2017 | 35.87 | 14.43 | 41.05 | 49.63 | 17.92 | 89.89 | 27.64 |
| 2018 | 36.60 | 14.42 | 29.38 | 42.05 | 22.64 | 87.05 | 27.92 |
| Mean Weighted Incidence | 35.85 | 12.01 | 34.95 | 42.63 | 17.27 | 222.42 | 27.43 |

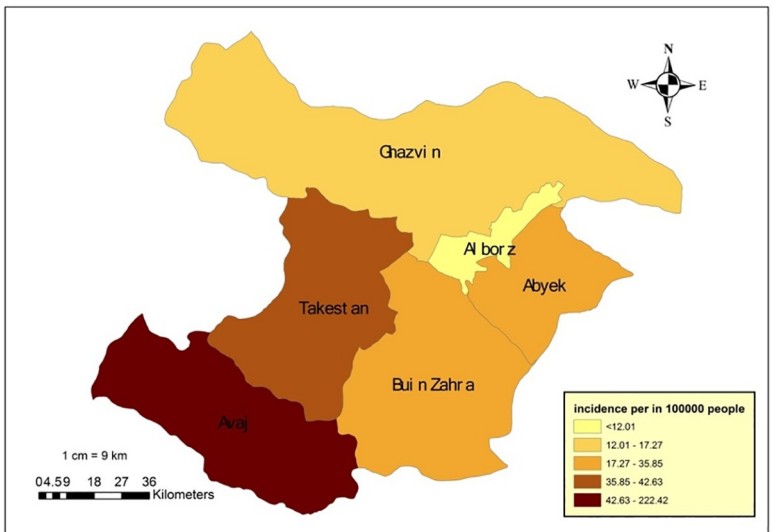

**Fig 1. Average incidence rate of brucellosis in Qazvin Provinces during 2010–2019.**

Also, the correlation between descriptive variables (climatic and non-climatic) and monthly brucellosis cases was presented (see Table 5). Fig 2 illustrates the time series graphs of the number of monthly brucellosis cases in Qazvin province. As it can be seen, the trends are nonlinear at provincial level, thus, classic time series methods do not efficiently work for these data. Correlation analysis was done to select appropriate inputs of modeling and significant ARIMA coefficients were considered as the inputs. The four artificial neural network methods (radial basis function and multilayer perceptron), support vector machine and random forest were fitted to each set. To compare the performance of the four models, the RMSE, MAE, MARE, and $R^2$ criteria were calculated for the training and test sets (See Table 6). Given these results, the RMSE, MARE and MAE values for the MLP method yielded smaller values compared to the other three ANN methods (RBF, RF, SVM). Furthermore, the $R^2$ value was closer to one in the MLP method compared to the other three ANN methods. Based on these findings, we may conclude that the MLP method performed better than the other three modeling and forecast methods for Qazvin province's monthly time series data sets–based on covariates and climatic parameters. The temporal changes of the observed cases of brucellosis and the values estimated by the four ANN methods, RA and SVM for the testing set are illustrated in Figs 3 and 4. As seen in the figure, the frequency of brucellosis has increased during the months of spring. This figure also demonstrates that the values forecasted by the MLP ANN method are better than the other three RBF, RF & SVM methods.

The remaining four methods' graphs are illustrated in Fig 4. The MLP method yielded smaller remnants, therefore, the performance of the MLP was better compared to the RBF, SVM and RF methods.

Moreover, Fig 5 depicts the observed values and estimates of (forecasted) brucellosis cases resulting from the four methods compared against each other using the scatter plot. As can be seen, all the points have fallen in the first one-fourth, which indicates that the estimated values are equal to the observed values. Moreover, the significance level of the fitted regression model was calculated for each of the four methods (MLP, RBF, RF and SVM) and was smaller than 0.001, which indicates the significance, validity and agreement between the observed and forecasted values in the four models. Given the results in Fig 5, the slope of the regression line was

**Table 4. Descriptive statistics of the monthly brucellosis cases in Qazvin Province.**

| Parameters | Statistics | Total data | Training set | Test Set |
|---|---|---|---|---|
| **Number of brucellosis cases** | Average (SD) | 31.3 (11.5) | 31.8 (12.1) | 29.3 (8.4) |
| | Min | 7.0 | 7.0 | 15.0 |
| | Max | 62.0 | 62.0 | 41.0 |
| **Rural Ratio** | Average (SD) | 2.0 (2.4) | 2.6 (2) | 1.3 (0.6) |
| | Min | 0.3 | 0.3 | 0.5 |
| | Max | 16.0 | 16.0 | 2.6 |
| **Average age** | Average (SD) | 38.4 (10.3) | 37.9 (11.3) | 40.7 (3.5) |
| | Min | 13.7 | 13.7 | 34.8 |
| | Max | 125.9 | 125.9 | 47.1 |
| **Male ratio** | Average (SD) | 2.2 (1.3) | 2.3 (1.4) | 2.2 (1.3) |
| | Min | 0.2 | 0.2 | 0.2 |
| | Max | 9.0 | 9.0 | 9.0 |
| **Ratio of ranchers** | Average (SD) | 0.3 (0.1) | 0.3 (0.1) | 0.3 (0.1) |
| | Min | 0.1 | 0.1 | 0.2 |
| | Max | 0.6 | 0.6 | 0.6 |
| **Ratio of Contact History** | Average (SD) | 3.6 (3.9) | 3.7 (4.2) | 3.0 (1.8) |
| | Min | 0.2 | 0.2 | 1.1 |
| | Max | 25.0 | 25.0 | 7.5 |
| **Ratio of history of consumption of unpasteurized dairy** | Average (SD) | 4.5 (4.3) | 4.1 (4.5) | 6.0 (3.3) |
| | Min | 0.6 | 0.6 | 1.0 |
| | Max | 27.0 | 27.0 | 14.0 |
| **Average temperature ($c^0$)-** | Average (SD) | 14.6 (9.1) | 14.9 (9.1) | 13.2 (8.9) |
| | Min | 0.9 | 0.9 | 2.7 |
| | Max | 30.3 | 28.2 | 30.3 |
| | Min | 0.00 | 0.00 | 0.00 |
| | Max | 89.3 | 89.3 | 83.6 |
| **Average wind speed** (Meter per Seconds) | Average (SD) | 1.9 (0.3) | 1.9 (0.4) | 2.0 (0.3) |
| | Min | 1.0 | 1.0 | 1.6 |
| | Max | 2.6 | 2.6 | 2.6 |
| **Maximum temperature ($c^0$)** | Average (SD) | 28.2 (9.6) | 28.7 (9.6) | 26.4 (9.9) |
| | Min | 12.5 | 12.5 | 14.0 |
| | Max | 42.5 | 42.5 | 41.7 |
| **Minimum temperature ($c^0$)** | Average (SD) | 2.0 (8.5) | 2.2 (8.6) | 0.8 (0.8) |
| | Min | -14.2 | -13.0 | -14.2 |
| | Max | 16.8 | 16.2 | 16.8 |

closer to 1 in the MLP model than in the other three models, which, once again, indicates the better performance of this method.

The significance of the variables used in the MLP have been shown in Fig 6; most climatic variables–particularly temperature and wind speed- influenced the number of brucellosis cases.

## Discussion

First, we will discuss the epidemiologic descriptive results of brucellosis in Qazvin province between the years 2010 and 2018. The incidence of the disease was on average 27.43 per 100000 person in the 9 years of the study, which, according to Zeynali & Shirzadi's classification falls in the highly infected regions (21–30 per 100000). We must however note that the

**Table 5. Correlation between descriptive statistics and the monthly brucellosis cases.**

| Variables | Pearson Correlation | P-value |
|---|---|---|
| Rural Ratio | 0.20 | 0.420 |
| Average age | -0.06 | 0.530 |
| Male ratio | -0.17 | 0.086 |
| Ratio of ranchers | 0.06 | 0.556 |
| Ratio of Contact History | 0.35 | <0.001 |
| Consumption of unpasteurized dairy | 0.33 | <0.001 |
| Average temperature (c$^0$)- | 0.33 | 0.001 |
| Wind speed (M/S) | 0.45 | <0.001 |
| Maximum temperature (c$^0$) | 0.31 | 0.001 |
| Minimum temperature (c$^0$) | 0.30 | 0.002 |
| Average wind speed (M.S) | 0.29 | 0.003 |

statistics reported are approximately 4 to 10 percent of the existent cases, a phenomenon that occurs even in developed countries. This happens due to the variety in clinical features, not visiting a physician when the clinical symptoms are mild, and incomplete registration and reporting [45–47]. Thus, we predict that the actual number of cases across the province are much higher than the official records. Shoraka et al reported the incidence of brucellosis in North Khorasan's Maneh and Samalghan counties at 25.2 and 38.6 per 100000 person, respectively, during the years 2008 and 2009 [48]. Farahani et al estimated this incidence rate in Arak at 60 cases per 100000 persons during 2001–2010 [49]. In our study, the most frequently affected age group was the 25–35-year-old age group. The disease was mostly prevalent in Qazvin's rural areas and in men, thus, it was mostly seen in rural males whose main occupation was ranching and who were in contact with livestock. The high percentage of the disease in this age group may be justified by their high person, heavy workload, and their direct contact with livestock. The finding that males are more commonly affected than females can be confirmed by Farahani et al's results in 2010 [50]. Another similar foreign study conducted by Donno et al in 2010 also indicated a higher percentage of brucellosis among males (66.2%) [51]. Although the results of Zeynalian et al's study in Esfahan state otherwise, i.e., the disease is more common among females [52]. In industrial nations, brucellosis has more often been reported in slaughterhouse workers and butchers [53]. Here, the most frequently affected occupations were ranchers–farmers (40.2%) and housewives (30.7%). The high prevalence among the latter group may be explained by the fact that rural housewives very often work alongside their spouses in ranching and farming and are therefore in contact with livestock and dairy

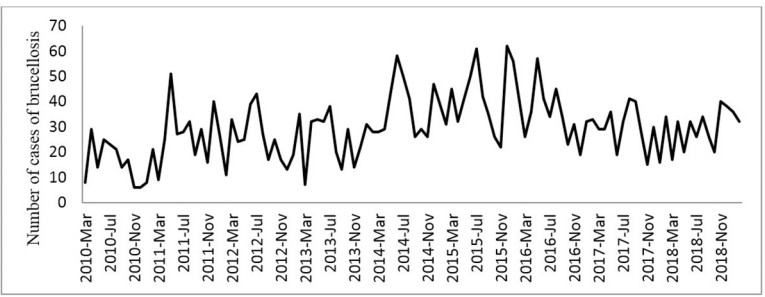

**Fig 2. Time series diagrams of the number of monthly brucellosis cases in Qazvin Province during the years 2010–2019.**

**Table 6. Evaluation of the prediction models over the test set.**

| Model | Evaluation criteria | | | |
|---|---|---|---|---|
| | **RMSE** | **MAE** | **MARE** | **R$^2$** |
| Multilayer Perceptron networks | 0.22 | 0.18 | 0.01 | 1.00 |
| Radial Basis Function | 8.46 | 7.49 | 0.31 | 0.03 |
| Random Forrest | 9.25 | 7.64 | 0.33 | 0.01 |
| Support Vector Machine | 8.21 | 6.58 | 0.29 | 0.08 |

products, thus being exposed to the risk of infection. In terms of occupation, the studies conducted by Medical Universities of Semnan, Kordestan, Birjand and Lorestan reported the highest prevalence of brucellosis among housewives [54–56]. Moreover, determining the seasonal prevalence of the disease indicated that it occurs most often in summer. Similarly, Esmail-nasab et al observed that brucellosis has higher prevalence during the months of May, June and July [57]. In 2011, Hamzavi et al studied the prevalence of the disease in Kermanshah, and found it to be more prevalent during the months of spring and in rural regions [58]. Elsewhere, in 2009, researchers observed that the prevalence of the disease reached 45 per 100000 person in East Azerbaijan and that it occurred more frequently during May and June [59]. Perhaps the higher prevalence of the disease during the warmer months of the year is due to the increased reproduction rate of livestock and greater contact with them. All the aforementioned results point towards one fact, that although many countries have been reported as brucellosis free, it is still prevalent in Iran, in spite of the considerable advancements made in its control; it is still a health problem, particularly in the western regions of the country and the outskirts of the Alborz mountain range, including the Qazvin province [60]. Our results indicated that, compared to its urban counterparts, the prevalence of the disease is higher in rural regions of Qazvin, a finding which underscores the necessity of laying greater focus on the control & prevention of brucellosis in this province and especially its rural areas. It seems that the habit of consumption of local dairy products–as an absolute must- and the ranching occupation and contact with livestock among the people of this region are the main reasons behind the relatively high prevalence of the disease. Given these findings, the residents of this province must be educated on the consumption of pasteurized dairy products. Another important point is the collaboration between the University of Medical Sciences and the provincial Central Veterinary Office to encourage ranchers to vaccinate their livestock, which is essential in significantly reducing the prevalence of the disease. Moreover, unawareness on the disease is another major reason why it cannot be controlled. The people and particularly rural residents and nomads who are exposed to the disease do not have adequate basic information about brucellosis, such that various studies across the country have shown low levels of awareness,

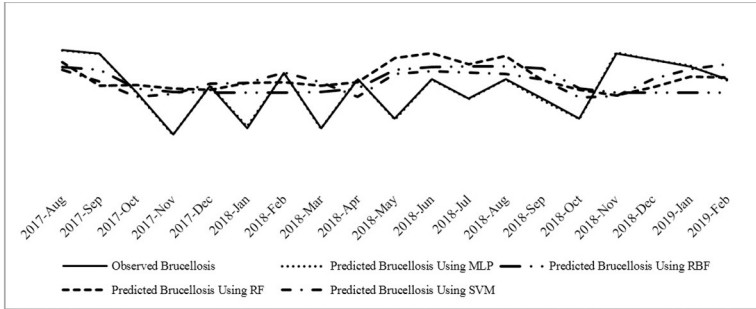

**Fig 3. Forecasted number of brucellosis cases obtained from MLP, RBF, RF and SVM time series.**

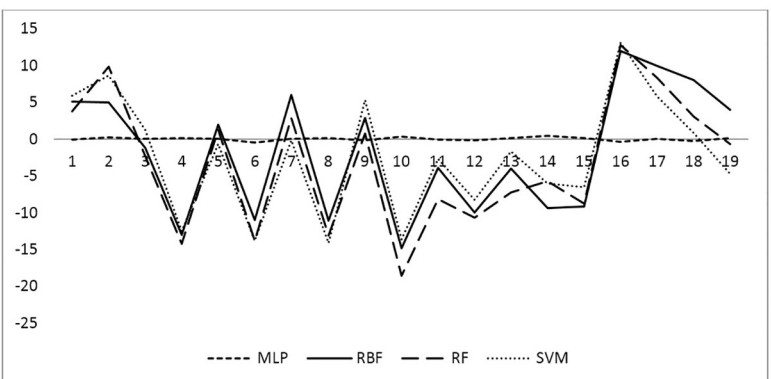

**Fig 4. Graph of the number of residuals obtained from fitting MLP, RBF, SVM, RF time series models.** SVM, RF.

knowledge and performance regarding this disease [61,62]. Furthermore, the highest incidence rates over the 9-year period were observed in Avaj (222.42 per 100000 person) and Takestan (42.63 per 100000 person), respectively; these rates are even higher than the provincial and national rates. Perhaps, the rural nature of these two counties, as well as their adjacency to infected provinces like Hamedan contribute to this high prevalence.

The second part of this study deals with the forecast results of brucellosis incidence by employing machine learning data analysis methods and their comparison in forecasting this rate in Qazvin province for the years 2010–2018. The precise and timely determination of infectious diseases' epidemics plays an important role in their control and prevention. This can be done through prevention strategies such as, sensitization and raising awareness among physicians on the rapid diagnosis of disease, correct treatment of patients, and delivery of health messages. Efficient statistical models of high precision can be useful tools for forecasting infectious disease outbreaks in the future [25]. The performance of statistical models is dependent on time series data, and there is no single model that can perform the best for all cases. Therefore, it is very important to assess and compare the performances of various statistical methods–particularly machine learning–based methods- as one can discover important and applicable information about their strengths and weaknesses [63], and acquire a better perspective on the utilization of better forecast models. Theory–based machine learning models have exhibited good performance in different fields of science, including time series analysis.

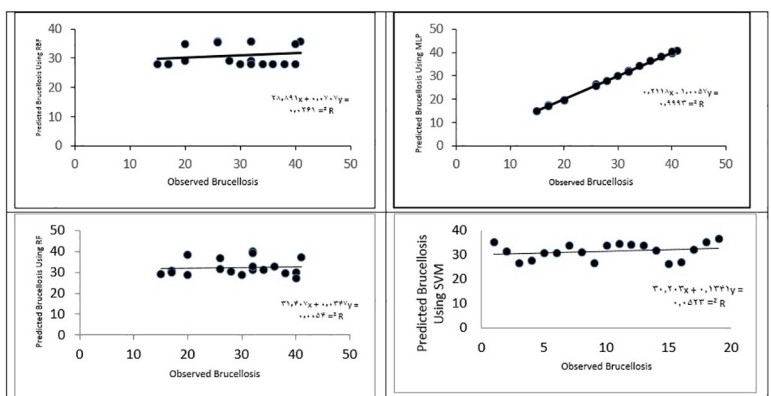

**Fig 5. Number of brucellosis cases observed and forecasted using four MLP, RBF, RF and SVM models in Qazvin Province.**

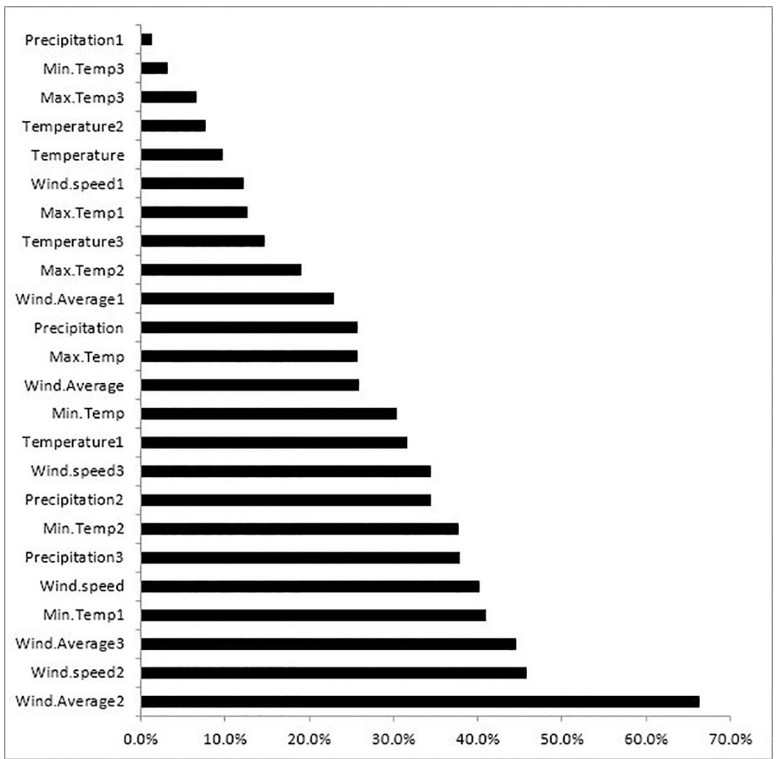

**Fig 6. Climatic importance chart for forecasting the number of monthly brucellosis cases in Qazvin Province.**

Based on literature, theory–based machine learning methods are effective and efficient in health systems. These methods are naturally beneficial forecast methods in time series analyses of endemic diseases, as they are capable of modeling nonlinear relations and data complexities.

In this study, that was conducted on human brucellosis cases of Qazvin province between 2010 and 2018, a total of 3194 patients were detected, upon which the accuracy of the four MLP, RBF, RF and SVM methods were modeled and compared. Based on our results, in comparison to the other three methods, the MLP method exhibited better performance in modeling the monthly changes of brucellosis, and estimated a trend closer to the one observed. The trends forecasted by the RBF, RF and SVM neural networks were very different from the one observed. The intercept of the values observed and those forecasted will lead to misleading planning in the health system [64]. The values of monthly brucellosis cases estimated by MLP showed very good agreement with the values observed. However, the values estimated by the other three methods (RBF, RF and SVM) did not show good agreement with the observed values. Since the differences between the observed and estimated values can lead to errors in the healthcare system, their disagreement is of utmost importance. Based on goodness of fit criteria (RMSE, MAE, MARE and R2), the graphs presenting the values forecasted by the MLP time series method were more powerful in forecasting the monthly cases of brucellosis, than those of the other three methods; the time series and non-series forecasted the number of brucellosis cases better than the other three models. The MLP's better performance, or, in other words, the smaller differences between its observed and forecasted values may be attributed to the utilization of the following in its modeling: historical data (values observed in the past 12 months) as forecasting variables in modeling, other influential parameters such as, mean temperature, minimum temperature, maximum temperature, precipitation, wind speed and

average wind speed, and other factors such as mean age, ratio of unpasteurized dairy product consumption, ratio of contact with livestock, males' ratio, ranchers' ratio, and rural ratio. The dissimilarity between the test and training data sets might severely affect a model and reduce its forecast power.

Like many other studies, we too concluded that MLP performs better in estimating the monthly cases of brucellosis [65–67]. However, our results do not conform to those observed by Bayram et al, wherein RBF–based monthly brucellosis time series analysis performed better than the combination of RBF and KNN. Therefore, our results showed that the MLP method can be effectively used in the monthly forecast of brucellosis. The MLP network is one of the most important artificial neural networks that are normally formed of multiple input layers and the input signal is distributed throughout the network in layers. Therefore, given its complicated structure it has better generalizability in forecasting the output variable. This task is undertaken through the identification of complicated temporal changes inside time series data [66]. Recently, studies have been conducted by various countries on the comparison of machine learning methods' performance aimed at forecasting health data. One of these studies is Zhang et.al study [19]. In this study, the classic methods of ARIMA and exponential smoothing were compared to SVM, where SVM exhibited a better performance. Guan et al compared the performance of neural networks with classic statistical models to forecast the incidence of hepatitis and showed that neural networks performed much better than classic statistical models [68]. In 2017, Oliveira et al also compared a few data mining methods, including the K-nearest neighbor and MLP networks. Of the methods employed, the MLP method was better than the rest [69]. Given that–to our knowledge- this study is the first in its kind in Qazvin province, we recommend future research studies to compare the performance of other data analysis methods in the field of brucellosis and/or other diseases in this province.

Another objective of this research was to study the detection of climatic and other risk factors influencing brucellosis. Given the bacterial nature of the cause of this disease, environmental factors such as, weather conditions and certain other influential factors can affect the occurrence of this disease. Thus, in addition to 1–12 month lag variables, here we used the following climatic data: average temperature, minimum temperature, maximum temperature, wind speed and average wind speed, precipitation, and other risk factors such as, month, year, season, mean age, gender ratio, ratio of unpasteurized dairy product consumption, and ratio of contact with livestock. Their impacts upon the disease were then examined using the aforementioned methods. Based on our results of the MLP model, we found that temperature and wind were directly related to the brucellosis incidence, and were the most influential factors compared to other climatic parameters. Qazvin province is located in a cold mountainous area with lowlands, thus, ranching thrives in this region. It appears that Qazvin's climatic conditions significantly affect the incidence of this disease, as when the temperature is suitable and the pastures are of good quality the livestock thrive and reproduce more. In other words, it may be said that when the average temperature is 15 degrees Centigrade, it can have the greatest effect among the climatic parameters one year later; meaning, this bacterium can remain alive in the environment for one year at this temperature in Qazvin. Undoubtedly, these bacteria live shorter during minimum and maximum temperatures, i.e. the incidence of the disease is lower during hot summers and cold winters, whereas, the moderate climate of Qazvin during these two seasons aggravates the disease. Wind reduces the incidence of this disease at high speeds, reason being that these bacteria live shorter in air. An increase in air pressure aggravates the disease, as higher pressure indicates air stability, and it seems that the disease flourishes in a relatively stable climate and suitable temperatures.

Finally, there are various statistical models in medical sciences that can predict disease behavior. Data Mining System for Infection Control Surveillance (DMSS) is one of a novel

approaches [70] for achieving the mentioned goal. Application of DMSS in health care data leads to the determination of rapid and accurate predicting outbreaks and it led to timely and appropriate health decisions of policymakers and epidemiologists.

One of the limitations of this study is the limited duration of the time series duration, which can partially reduce the forecast model's performance. Another limitation is the lack of comparison between machine learning based–statistical methods and classic methods.

## Conclusion

Based on our results, the MLP artificial neural network model can be used for detecting changes in behavior of human brucellosis cases over time and based on changes in climatic parameters. Most climatic parameters were influential in the incidence of the disease, and the most influential one was temperature. Further studies on the practical application of time series models and detection of the best model for the control and prevention of brucellosis are warranted.

## Supporting information

**S1 Dataset.**
(XLS)

## Acknowledgments

We would hereby like to extend our gratitude to Qazvin University of Medical Sciences' Head of Department of Disease Prevention and Control, Dr. Shiva Leghaee and her colleagues who helped in data extraction.

## Author Contributions

**Conceptualization:** Hadi Bagheri, Leili Tapak, Zahra Hosseinkhani, Hamidreza Najari, Safdar Karimi, Zahra Cheraghi.

**Data curation:** Hadi Bagheri, Zahra Cheraghi.

**Formal analysis:** Hadi Bagheri, Leili Tapak, Zahra Cheraghi.

**Funding acquisition:** Hadi Bagheri, Zahra Cheraghi.

**Investigation:** Hadi Bagheri, Zahra Cheraghi.

**Methodology:** Hadi Bagheri, Manoochehr Karami, Zahra Cheraghi.

**Project administration:** Hadi Bagheri, Zahra Cheraghi.

**Resources:** Hadi Bagheri, Zahra Cheraghi.

**Software:** Hadi Bagheri, Leili Tapak, Zahra Cheraghi.

**Supervision:** Hadi Bagheri, Zahra Cheraghi.

**Validation:** Hadi Bagheri, Zahra Cheraghi.

**Visualization:** Hadi Bagheri, Zahra Cheraghi.

**Writing – original draft:** Hadi Bagheri, Zahra Cheraghi.

**Writing – review & editing:** Hadi Bagheri, Zahra Cheraghi.

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
