## [Decision Letter · Decision Letter 0]

25 Feb 2020

PONE-D-20-00374

Forecasting the monthly incidence rate of brucellosis in west of Iran using time series and data mining from 2010 to 2019

PLOS ONE

Dear Dr Cheraghi,

Thank you for submitting your manuscript to PLOS ONE. After careful consideration, we feel that it has merit but does not fully meet PLOS ONE’s publication criteria as it currently stands. Therefore, we invite you to submit a revised version of the manuscript that addresses the points raised during the review process.

Reviewers' comments are listed to perform the required changes before acceptance of your work. 

We would appreciate receiving your revised manuscript by Apr 10 2020 11:59PM. To enhance the reproducibility of your results, we recommend that if applicable you deposit your laboratory protocols in protocols.io, where a protocol can be assigned its own identifier (DOI) such that it can be cited independently in the future. For instructions see: http://journals.plos.org/plosone/s/submission-guidelines#loc-laboratory-protocols

We look forward to receiving your revised manuscript.

Kind regards,

Esteban Tlelo-Cuautle, Ph.D

Academic Editor

PLOS ONE

Additional Editor Comments (if provided):

You are encouraged to attend reviewers' comments to improve the impact of your work.

Journal Requirements:

2. Thank you for stating the following in the Cover letter of your manuscript:

"This study was funded by the Vice-Chancellor of Research and Technology of Hamadan

University of Medical Sciences. The funders had no role in study design, data collection and analysis,

decision to publish, or preparation of the manuscript."

"None."

"None declared."

5. Please amend your authorship list in your manuscript file to include author Hadi Bagheri, Leili Tapak, Manoochehr Karami, Zahra Hasankhani, Hamidreza Najari, Safdar Karimi

6. Please amend your list of authors on the manuscript to ensure that each author is linked to an affiliation. Authors’ affiliations should reflect the institution where the work was done (if authors moved subsequently, you can also list the new affiliation stating “current affiliation:….” as necessary).

7. We note you have included a table to which you do not refer in the text of your manuscript. Please ensure that you refer to Table 5 in your text; if accepted, production will need this reference to link the reader to the Table.

8. Please include a copy of Table 10 which you refer to in your text on page 7.

Reviewers' comments:

Reviewer's Responses to Questions

**Comments to the Author**

1. Is the manuscript technically sound, and do the data support the conclusions?

Reviewer #1: Yes

Reviewer #2: Yes

2. Has the statistical analysis been performed appropriately and rigorously? 

Reviewer #1: Yes

Reviewer #2: Yes

3. Have the authors made all data underlying the findings in their manuscript fully available?

Reviewer #1: Yes

Reviewer #2: Yes

4. Is the manuscript presented in an intelligible fashion and written in standard English?

Reviewer #1: Yes

Reviewer #2: Yes

5. Review Comments to the Author

Reviewer #1: Dear Editor

Please find the comments below.

Bests,

Comments for the authors

The authors compared the performance of four machine learning methods in forecasting human brucellosis. Generally the subject is interesting and the manuscript is well written. In my concern, there are some minor issues with the manuscript as follows:

Are there other disease that could look like the case definition? How sensitive and specific to Brucellosis is that definition.

Do the patients have to have all of the signs at one time?

Do the authors mean random forest by “random accumulation”? Please correct them as random accumulation is not the usual term.

According to golden rules of reporting the numbers (BMJ publication) the numbers under 10, must be presented in letters!

The majority of the references (References: 5, 6, 20 and 22) are not up to date.

All formulas must be numbered.

Reviewer #2: Overall, I think this manuscript is well written. Just a few suggestions for the result section.

1. Table 1 and Table 2, add p values to compare the incidence between each characteristics.

2. Add a time series correlation matrix or plot to show the correlation between the characteristics in Table 4 and the time series of brucellosis.

3 Explain in detail how you optimize the parameters in your four machine learning models. For example, how did you choose the number of hidden layers for neural network. Please also list those parameters for the other models, and explain how did you determine these parameters.

4 In the discussion section, please include more discussion how to apply these models in real infectious disease surveillance.

6. PLOS authors have the option to publish the peer review history of their article (what does this mean?). If published, this will include your full peer review and any attached files.

Reviewer #1: No

Reviewer #2: Yes: Xingyu Zhang

---

## [Author Response · Author response to Decision Letter 0]

30 Mar 2020

Dear Editor,

Thank you for your constructive and valuable comments. Below we have provided a point-by-point response to all the comments. 

Comments to the Author

The authors compared the performance of four machine learning methods in forecasting human brucellosis. Generally, the subject is interesting and the manuscript is well written. In my concern, there are some minor issues with the manuscript as follows:

Are there other disease that could look like the case definition? How sensitive and specific to Brucellosis is that definition.

No, there aren't. The clinical – epidemiological symptoms of brucellosis were considered disease cases including : fever, myalgia and para-clinical symptoms are very non-specific and it may be like to several disease, but the definition of definitive case in our study was the positive results of two routine lab tests for brucellosis) including, Wright’s (diagnostic test for brucellosis; values greater than 1.8 indicate presence of infection) and 2ME (Mercaptoethanol Brucella agglutination test) (brucellosis confirmatory test, which if greater than or equal to 1.4 is indicative of the presence of infection. Please refer to Page 13th lines: 132-35

Do the patients have to have all of the signs at one time?

Not necessarily, because this is the prospective cohort studies, that may be people affect the brucellosis in various time points.

Do the authors mean random forest by “random accumulation”? Please correct them as random accumulation is not the usual term.

Thanks revised.

According to golden rules of reporting the numbers (BMJ publication) the numbers under 10, must be presented in letters!

Thanks revised.

The majority of the references (References: 5, 6, 20 and 22) are not up to date.

Thanks, revised.

All formulas must be numbered. Thanks, revised. Pages 5-6

Thanks revised.

Reviewer #2: Overall, I think this manuscript is well written. Just a few suggestions for the result section.

1. Table 1 and Table 2, add p values to compare the incidence between each characteristics.

Thanks, revised. Please see table 1 and 2

2. Add a time series correlation matrix or plot to show the correlation between the characteristics in Table 4 and the time series of brucellosis.

The table added. Page 22 

3 Explain in detail how you optimize the parameters in your four machine learning models. For example, how did you choose the number of hidden layers for neural network. Please also list those parameters for the other models, and explain how did you determine these parameters.

Edited. Page 8th lines 198-210 as follow as:

To implement the models, variables in Table 1 as well as climatic variables of wind speed (m/s) and temperature (Centigrade) were used as predictors and the numbers of brucellosis cases observed was used as the output….

4 In the discussion section, please include more discussion how to apply these models in real infectious disease surveillance.

Thanks, revised, as follow as: Page 13th Lines: 372-75

---

## [Decision Letter · Decision Letter 1]

24 Apr 2020

Forecasting the monthly incidence rate of brucellosis in west of Iran using time series and data mining from 2010 to 2019

PONE-D-20-00374R1

Dear Dr. Cheraghi,

We are pleased to inform you that your manuscript has been judged scientifically suitable for publication and will be formally accepted for publication once it complies with all outstanding technical requirements.

With kind regards,

Esteban Tlelo-Cuautle, Ph.D

Academic Editor

PLOS ONE

Additional Editor Comments (optional):

The updated manuscript is fine to be accepted

Reviewers' comments:

Reviewer's Responses to Questions

**Comments to the Author**

1. If the authors have adequately addressed your comments raised in a previous round of review and you feel that this manuscript is now acceptable for publication, you may indicate that here to bypass the “Comments to the Author” section, enter your conflict of interest statement in the “Confidential to Editor” section, and submit your "Accept" recommendation.

Reviewer #1: (No Response)

Reviewer #2: All comments have been addressed

2. Is the manuscript technically sound, and do the data support the conclusions?

Reviewer #1: (No Response)

Reviewer #2: Yes

3. Has the statistical analysis been performed appropriately and rigorously? 

Reviewer #1: (No Response)

Reviewer #2: Yes

4. Have the authors made all data underlying the findings in their manuscript fully available?

Reviewer #1: (No Response)

Reviewer #2: Yes

5. Is the manuscript presented in an intelligible fashion and written in standard English?

Reviewer #1: (No Response)

Reviewer #2: Yes

6. Review Comments to the Author

Reviewer #1: Dear Editor

There is no comments for this manuscript and the authors provided approperiate answers.

The decision is to accept.

Reviewer #2: (No Response)

7. PLOS authors have the option to publish the peer review history of their article (what does this mean?). If published, this will include your full peer review and any attached files.

Reviewer #1: No

Reviewer #2: No

---

## [Editor Report · Acceptance letter]

28 Apr 2020

PONE-D-20-00374R1 

Forecasting the monthly incidence rate of brucellosis in west of Iran using time series and data mining from 2010 to 2019 

Dear Dr. Cheraghi:

I am pleased to inform you that your manuscript has been deemed suitable for publication in PLOS ONE. Congratulations! Your manuscript is now with our production department. 

With kind regards,

on behalf of

Dr. Esteban Tlelo-Cuautle 

Academic Editor

PLOS ONE